# Metabolomic-Based Approaches for Endometrial Cancer Diagnosis and Prognosis: A Review

**DOI:** 10.3390/cancers16010185

**Published:** 2023-12-29

**Authors:** Manel Albertí-Valls, Cristina Megino-Luque, Anna Macià, Sònia Gatius, Xavier Matias-Guiu, Núria Eritja

**Affiliations:** 1Oncologic Pathology Group, Biomedical Research Institute of Lleida (IRBLleida), University of Lleida, Av. Rovira Roure 80, 25198 Lleida, Spain; crismegino.cm@gmail.com (C.M.-L.); amacia@irblleida.cat (A.M.); sgatius.lleida.ics@gencat.cat (S.G.); fjmatiasguiu.lleida.ics@gencat.cat (X.M.-G.); 2Department of Medicine, Division of Hematology and Oncology, The Tisch Cancer Institute, Icahn School of Medicine at Mount Sinai, New York, NY 10029, USA; 3Centro de Investigación Biomédica en Red de Cáncer (CIBERONC); 4Laboratory of Precision Medicine, Oncobell Program, Bellvitge Biomedical Research Institute (IDIBELL), Department of Pathology, Hospital de Bellvitge, Gran via de l’Hospitalet 199, 08908 Barcelona, Spain

**Keywords:** endometrial cancer, metabolomics, biomarker discovery, metabolic profile

## Abstract

**Simple Summary:**

Endometrial cancer, the most common gynecological malignancy in developed countries, poses a growing challenge with rising incidence and mortality rates. This review explores how metabolomic technology can offer valuable insights into the molecular aspects of the disease. By identifying new possible metabolite biomarkers, it has the potential to improve the accuracy of diagnosis, prognosis, and monitoring, thereby revolutionizing the management of endometrial cancer.

**Abstract:**

Endometrial cancer, the most prevalent gynecological malignancy in developed countries, is experiencing a sustained rise in both its incidence and mortality rates, primarily attributed to extended life expectancy and lifestyle factors. Currently, the absence of precise diagnostic tools hampers the effective management of the expanding population of women at risk of developing this disease. Furthermore, patients diagnosed with endometrial cancer require precise risk stratification to align with optimal treatment planning. Metabolomics technology offers a unique insight into the molecular landscape of endometrial cancer, providing a promising approach to address these unmet needs. This comprehensive literature review initiates with an overview of metabolomic technologies and their intrinsic workflow components, aiming to establish a fundamental understanding for the readers. Subsequently, a detailed exploration of the existing body of research is undertaken with the objective of identifying metabolite biomarkers capable of enhancing current strategies for endometrial cancer diagnosis, prognosis, and recurrence monitoring. Metabolomics holds vast potential to revolutionize the management of endometrial cancer by providing accuracy and valuable insights into crucial aspects.

## 1. Methods

This review adheres to the guidelines outlined in the Preferred Reporting Items for Systematic Reviews and Meta-Analyses (PRISMA) [1]. A-V.M. and E.N. conducted the study selection using two major databases, PubMed and Scopus, before December 2023. Various combinations of terms such as “endometrial cancer”, “metabolomics”, “endometrial cancer biomarkers”, “metabolomics for biomarker discovery”, and “data analysis for metabolomics” were employed in the search. Given the interdisciplinary nature of this study, a wide range of data sources, encompassing peer-reviewed articles such as reviews, systematic reviews, and investigative papers, as well as book chapters were considered. Articles were considered eligible for the review if they met the following inclusion criteria: (a) gave relevant information on endometrial cancer, (b) gave relevant information on metabolomics, data analysis, or biomarker discovery (c) gave relevant information on endometrial cancer biomarker discovery for either diagnosis or prognosis. Exclusion criteria were: (i) case reports, editorials, unpublished studies, (ii) in silico analysis only, (iii) data not fully written in English.

## 2. Endometrial Cancer

### 2.1. Epidemiology

Endometrial cancer (EC) is the most prevalent gynecological malignancy across developed countries [2]. The incidence of EC is steadily increasing, together with disease-associated mortality, and has become the sixth most common female cancer with 471,336 cases and 89,000 deaths in 2020 [3,4]. Three factors can be directly linked to the EC increase: the demographic shift towards an aging population, continued fertility decline, and the growing prevalence of obesity [5,6].

### 2.2. Diagnosis

Postmenopausal bleeding (PMB) is a prevalent symptom of EC and accounts for around two-thirds of all gynecological visits among perimenopausal and postmenopausal women. While postmenopausal bleeding is considered a red-flag symptom of EC and is reported in 90% of women with EC, the likelihood of an EC diagnosis in women with PMB is only between 5% to 10% [7,8,9].

Diagnosing EC typically involves a standard work-up that may include transvaginal ultrasound, which can be followed by endometrial biopsy, or dilatation and curettage, often accompanied by hysteroscopy. These procedures can be costly, invasive, and painful for patients, with a risk of life-threatening uterine perforation and other complications. Thus, the development of a safer, more cost-effective alternative to the current standard diagnostic practice is urgent. In this context, myriad biomarkers from distinct biological origins (lipids, sugars, nucleic acids, proteins, etc.) have been studied but only a few, namely, CA-125 or HE4, have been successfully developed [10]. However, these methodologies are not yet considered as the standard for EC diagnosis because there is a lack of definitive clinical evidence to support their extended use [11,12,13,14].

### 2.3. Prognosis

Most patients with EC are diagnosed at early stages of disease. Overall, the five-year survival rate for endometrial cancer stands at an encouraging 81%. The outlook becomes more favorable when the cancer is detected and treated in its early stages. For patients who receive treatment before the cancer progresses beyond stage II, and if a hysterectomy is effectively performed, the survival rate skyrockets to an impressive 95% [15]. In young women diagnosed with early-stage EC with no invasion of the myometrium, it should be a priority to preserve fertility. Metabolomic screening presents an opportunity to diagnose and categorize those women, offering valuable insights into the suitability of fertility-preserving treatments for specific patients [16]. Unfortunately, prognosis takes a dramatic downturn when EC advances to stage III. At this juncture, the five-year survival rate drops to 17%. This contrast in survival rates highlights the critical role played by routine screenings, quick diagnostics, and effective treatment. Other than spread or tumor grade, there are other factors that are taken into account when performing a prognosis: risk of nodal involvement and muscle invasion, which have a proven influence in recurrence, and, finally, hormone receptor status [8].

In conclusion, the statistics surrounding endometrial cancer survival rates underscore the significance of early detection and appropriate treatment options. While an 81% overall survival rate is encouraging, it is essential to strive for early diagnosis to maximize the chances of a positive outcome, with a remarkable 95% survival rate attainable in cases where treatment commences before the cancer progresses.

### 2.4. EC Types: Histological and Molecular Classification

Defining the histological tumor type is a very relevant prognostic factor in EC. Traditionally, ECs were classified either as type I or endometrioid carcinoma or type II or non-endometrioid endometrial carcinoma. This dualistic classification has proven ineffective due to the overlapping characteristics at the clinicopathological and molecular levels [17,18]. The 5th edition of WHO Classification of Female Genital Tumors [19] has generated a new classification method based on 8 groups: (i) Endometrioid carcinoma (EEC); (ii) serous carcinoma (SC); (iii) clear cell carcinoma (CCC); (iv) mixed carcinomas (MC); (v) undifferentiated carcinoma (UC); (vi) carcinosarcoma (CS); (vii) other unusual types, such as mesonephric-like; and (viii) gastrointestinal mucinous type carcinomas. All of these different histological tumor types exhibit different precursor lesions, molecular features, microscopic appearance, and clinical progression [19]. Moreover, there are additional histopathological characteristics that can be analyzed to determine prognosis or risk stratification of ECs such as tumor grade and Federation of Gynecology and Obstetrics (FIGO) stage [20,21].

In 2013, the Cancer Genome Atlas (TCGA) Research Network revolutionized the approach to EC classification [22]. Using a combination of whole genome or exome sequencing, microsatellite instability (MSI) assays, and copy number analysis, ECs were grouped into 4 distinct molecular subtypes. Subsequent research has revealed that cost-effective immunohistochemical and molecular tests can function as substitutes for the intricate and expensive analyses conducted by TCGA, providing a more economically viable alternative [23,24]. The four EC molecular subtypes defined by the TCGA include: (i) the POLE ultra-mutated group comprising tumors with POLE exonuclease domain mutations that has the best prognosis, (ii) the MSI or mismatch repair deficient (MMRd) hypermutated group with intermediate prognosis, (iii) the p53 abnormal tumor group (also named “copy number high”) with the worst prognosis, (iv) and, finally, the group characterized by a low number of somatic copy number alterations (“also known as copy number low”), which presents good to intermediate prognosis.

## 3. Metabolomics: A Tool against Cancer

Over the past four decades, solid advancements in technology and computational tools within the field of metabolomics have greatly increased the accessibility of these technologies. These developments have paved the way for the creation of personalized metabolic profiling, which, when combined with personalized genomics, form the cornerstone of personalized medicine. These emerging metabolomic platforms have opened new possibilities in cancer screening, diagnosis, and treatment [25].

The metabolome is the total number of metabolites present within an organism. Since metabolites are both intermediate and downstream molecules of the genome, transcriptome, and proteome, the metabolome serves as a comprehensive representation of the human phenotype in both health and disease, effectively summarizing the findings of other ‘*omic*’ technologies. It is considered that metabolites associated with cancer may arise from two distinct sources: Firstly, as by-products of cellular processes triggered by neoplastic transformation and cellular proliferation and, secondly, due to the body’s immunological response to the disease [26,27].

In metabolomics analysis, two distinct approaches are employed, each serving different objectives: Nontargeted metabolomics and targeted metabolomics. Nontargeted metabolomics, also referred to as profiling metabolomics, takes a hypothesis-free approach by striving to detect as many metabolites as possible in a given sample. On the other hand, targeted metabolomics aims to precisely quantify the absolute concentrations of a predefined set of metabolites and is therefore, a hypothesis-driven approach [28]. While untargeted metabolomics are very suitable for the discovery phase of biomarker research, targeted approaches are applied for verification and validation of such biomarkers, hence, the combination of both approaches is needed to complete the discovery and validation of a biomarker [26].

### 3.1. Metabolomics Workflow

A metabolomics assay for either cancer diagnosis or prognosis involves a systematic workflow to analyze and identify changes in the metabolome of individuals to detect cancer-related biomarkers. This workflow typically consists of several key steps that are all equal in relevance (Figure 1).

#### 3.1.1. Sample Selection and Handling

The careful selection of patients is very relevant when conducting biomarker discovery or validation studies to mitigate selection bias. This bias arises when study participants are chosen in a manner that may not accurately represent the intended population for analysis [29]. To minimize selection bias, it is essential that both cases and controls closely resemble each other in all aspects except for the condition of interest. For example, for diagnostic and prognostic biomarker studies related to EC, controls should include women undergoing investigation for PMB who do not have EC. Whenever possible, controls should be matched to cases based on demographic factors such as age, ethnicity, BMI, comorbidities, and lifestyle factors [30]. Cases should be confirmed to have EC through histology to reduce misclassification bias. Additionally, sample size considerations are closely linked to statistical power and are vital for the reproducibility of study findings [28,31].

In the field of EC metabolomics, five main categories of samples are typically utilized, including blood, serum, urine, tumor tissue, and uterine aspirates (UA). Among these, blood, and serum, often collectively referred to as blood-based samples, are particularly favored as sources of biomarkers. This preference arises from their high accessibility and the ability to obtain repeated measurements over short time intervals. In contrast, collecting tumor tissue or UA is more invasive and distressing [32,33,34]. Although blood biomarkers are more cost and time-effective than tumor or UA samples, their immediate practical applicability in clinical settings is relatively uncertain. The main limitations stem from the challenges of measuring and standardizing thresholds, as well as their lack of specificity and sensitivity. On the other hand, tumor tissue sampling, while less palatable and conforming to patients’ preferences, serves as the gold standard for diagnosis by pathologists worldwide and since extraction comes from the primary site of cancer, the amount of confounding variables is reduced [33,35,36,37].

An alternative biomarker source is uterine aspiration, a minimally invasive medical procedure that employs vacuum aspiration to safely remove contents of the uterus. This procedure stands as an important intermediary step bridging the delicate balance between patient comfort and providing robust biomarkers. With its growing importance in the field of medical research, there is a body of evidence that indicates the utility of UA as a means to develop innovative biomarkers for EC. Numerous reports have already harnessed the potential of UA in their efforts to uncover novel and more precise biomarkers associated with EC [38,39,40].

A final source of metabolomics-based biomarkers is urine. Urine is an advantageous source of metabolic-based biomarkers due to its non-invasive nature and ease of collection without any associated risks. Urinary metabolites can arise from systemic compounds that undergo chemical alterations and are subsequently excreted in urine, or from the possible presence of tumor-derived metabolites released into the lower genital tract and contaminating the urine. Nevertheless, only a limited number of urinary metabolomic investigations have been conducted in the context of EC and there is still uncertainty regarding its clinical utility [41,42].

Maintaining the integrity of samples is of utmost importance to discover reliable metabolomic biomarkers. Pre-analytical factors related to sample collection, storage, transportation, and processing can introduce false signals into clinical samples, leading to erroneous positive results [43]. Therefore, it is essential to handle samples with care and consistency to ensure meaningful outcomes. Standard operating procedures should be in place, including quality control checks at every step of the analytical process. Exposure of clinical samples to unfavorable conditions that could lead to significant metabolite degradation should be avoided. For instance, sample preparation techniques should include temperature and pH regulation to prevent alterations in sample metabolites. It is also recommended to encourage effective sample storage practices, such as storing samples in multiple aliquots, to minimize the introduction of artifacts from multiple freeze-thaw cycles, which can affect study results [44,45].

#### 3.1.2. Sample Processing

Metabolomic profiling is commonly performed on two different platforms: Nuclear magnetic resonance spectroscopy (NMR) and Mass spectrometry (MS). However, there are alternative platforms like metabolic flux analysis and vibrational spectroscopy [46]. NMR spectroscopy is a powerful analytical technique used to examine the composition of samples by harnessing the unique electromagnetic properties of metabolites within a given sample when subjected to a magnetic field. Each metabolite within a sample emits its distinct NMR signal, which is sensitive to its molecular structure and chemical environment. The NMR platform offers quick results and the chance to safeguard samples for future investigations [47].

There are a number of chromatography platforms that can be coupled to MS for molecule separation including: Ion-mobility-MS (IM-MS), direct injection-MS (DI-MS), liquid chromatography (LC-MS), capillary electrophoresis-MS (CE-MS), and gas chromatography-MS (GC-MS). All of them provide vast amounts of chemical information and are more suitable for quantitative screenings. However, chromatography platforms are more expensive and time-consuming than NMR. The two most popular approaches are LC-MS and GC-MS. LC-MS is a moderately high-throughput method that allows to separate an indefinite number of mixed compounds in a liquid phase using a stationary phase. It obtains good molecular resolution and presents a rather simple sample preparation. LC-MS is ideal for studying intricate biological samples and has emerged as the most commonly used technique in metabolite profiling of biological tissue since it can process high molecular weight, polar, and thermally liable molecules, unlike GC-MS. A further optimization is the ultra-high pressure liquid chromatography, which is capable of producing the best resolution and has a higher peak capacity by packing samples in <2 µm units and employing very high pressure to improve the mobile phase speed. On the other hand, GC-MS is commonly used for the analysis of small molecular substances that are thermally stable. It relies on sample vaporization at >300 °C, which obtains astounding resolution and very high reproducibility. However, it’s a rather expensive technique with intricate sample preparation that requires derivatization of the original analytes [48,49].

Following the separation of molecules based on their retention time through chromatography, the next step involves their analysis using either a single-configuration mass analyzer (MS) or tandem (MS/MS). The most commonly used mass analyzers are the Quadrupole (Q), Quadrupole ion trap (QIT), Time-of-flight (TOF), and Orbitrap (OT). In this procedure, molecules are sorted based on their mass-to-charge ratio by utilizing either an electric or magnetic field to control the movement of ions generated from a target analyte. These ions are subsequently directed through a detector, which quantifies the abundance of ions at each mass-to-charge ratio. The acquired data is then subjected to analysis and compared with existing mass spectral databases to determine the molecular constituents. The exceptional sensitivity of mass analyzer makes it suitable for both targeted and non-targeted analyses. However, it is highly dependent on specific experimental conditions and instrument settings; therefore, sample preparation and handling become critical steps that determine the outcome of the analysis. Mass analyzer offers the chance to perform large-scale clinical studies with minimal sample volume due to its high sensitivity, but it is laborious and consumes the sample. For these reasons, thorough application of a suitable platform for each goal is essential to correctly perform metabolomics analysis [6,50].

#### 3.1.3. Data Analysis

Adequate data analysis is equally important as correct sample preparation and handling. The choice of statistical analysis is particularly important given the number of features that are simultaneously analyzed and the impact of confounding variables. The risk of a false positive test result is very high in this type of assay; therefore, certain statistical tools need to be applied. Confounding variables derived from demographic variability and exogenous metabolite variability can be managed via group stratification, exclusion of metabolite markers, and application of multivariate statistics [51,52]. Raw data is processed to remove noise, correct for instrument variability, and align spectra accurately. Data preprocessing steps may include peak picking, noise reduction, baseline correction, and normalization. Processed data is subjected to statistical and bioinformatic analyses. Multivariate statistical techniques, such as principal component analysis (PCA) and partial least squares-discriminant analysis (PLS-DA), are commonly used to identify patterns and differences between cancer and control groups. Univariate statistical tests may be employed to identify individual metabolites that are significantly altered in cancer samples. Additionally, hierarchy and classification analysis such as heatmaps and random forests are applied to visualize the degree of separation between study groups [53,54,55,56]. Metabolites that exhibit significant differences between cancer and control groups are considered potential biomarkers. To gain insights into the underlying biological mechanisms, pathway analysis is conducted to identify metabolic pathways that are perturbed in cancer as this can help in understanding the metabolic alterations associated with the disease.

#### 3.1.4. Model Development and Clinical Validation

Artificial Intelligence (AI) has gained significant attention in the last years, both as a conceptual framework and a robust research field. It offers a multitude of applications for comprehending structures and trends within vast datasets produced by modern high-throughput experiments [51].

Machine Learning (ML) algorithms are fundamentally rooted in their ability to construct mathematical models from a set of sample data [57,58]. Typically, a dataset used to develop a machine learning model is split into two subsets: a training subset, and a testing subset. The initial learning process relies on an ample supply of data, providing the ML algorithm with numerous opportunities to learn and refine the model. The training dataset guides the algorithms to make predictions without the need for explicit programming. The final step is to assess the model’s performance on an independent dataset containing previously unseen data [59].

The union of AI and ML opens up an extensive realm of possibilities. ML, specifically, is instrumental in crafting models capable of handling massive datasets and resolving intricate problems. However, in comparison to other biomedical and life science areas, such as neuroscience and genomics, the incorporation of AI into metabolomics research has lagged. One primary reason for the slower integration of AI in the metabolism field is the scarcity of high-quality datasets, which are fundamental for the successful implementation of ML algorithms and platforms.

Nevertheless, there are already several reports that utilize this technology to develop EC biomarkers. For instance, in 2022, Troisi and colleagues successfully developed a machine learning model that obtained 96% accuracy in EC diagnosis [60]. In another study, Houri and colleagues generated a machine learning model to predict EC recurrence. Employing a multifactorial approach, their model showed a specificity of 55% and a sensitivity of 98%, with an AUC (area under the curve) of 0.84 [61].

Next, biomarkers and models are validated in larger clinical cohorts to assess their clinical utility and diagnostic accuracy. Subsequently, results are reported, and interpretations are made based on the identified biomarkers and predictive models. Finally, clinical trials and studies are conducted to evaluate the efficacy of metabolomics-based cancer diagnosis in real-world settings. Consequently, clinical reports are generated to aid in cancer diagnosis and treatment decisions. Additionally, metabolomics data can be integrated with genomic, proteomic, and clinical data to provide a comprehensive understanding of cancer biology.

## 4. Metabolomics: Applications in EC Diagnosis and Prognosis

### 4.1. Metabolomics for EC Diagnosis

An optimal EC diagnostic biomarker should possess the capability to identify ECs at various grades and stages in women experiencing symptoms and to screen asymptomatic high-risk individuals, with minimal occurrence of false positives or false negatives. In recent years, multiple groups have described various metabolite signatures that aim to accurately diagnose EC (summarized in Table 1).

The most commonly dysregulated metabolic pathways in EC are lipid and glycolysis-related pathways. Consequently, there are multiple reports that have identified phosphocholines (PC), acylcholines, carnitines, and other lipid by-products as promising diagnostic biomarkers. For example, a study developed by Knific and colleagues, employed 126 plasma samples (61 patients with endometrial cancer and 65 control patients) and identified three phosphatidylcholines (PC C40:1, PC C42:0 and PC C44:5) that are decreased in EC patients. Furthermore, they composed a diagnostic model that is defined as the ratio between acylcarnitine C16 and phosphatidylcholine PC C40:1, the ratio between proline and tyrosine, and the ratio between the two phosphatidylcholines PC C42:0 and PC C44:5; which provided a sensitivity of 85.25%, a specificity of 69.23%, and an AUC of 0.837 [62]. Moreover, these biomarkers have proven to be dysregulated in other sample types such as cervicovaginal fluids and endometrial tumor tissues [63,64]. In another report, Cheng and colleagues designed a unique lipid biomarker panel that gathers lipidomic and transcriptomic data. They developed a machine learning model using a total of 78 patients: 38 samples were used as the discovery set and 40 samples as the validation set. The lipidomic study unveiled an increase in various lipid species (isolithocholic acid, TG (16:0) among others) and a decrease in certain carnitines (Carnitine C9:0 and Carnitine C10:1-OH). Additionally, pathway enrichment analysis consistently demonstrated disturbances in sphingolipid and glycerophospholipid metabolism. Consequently, a lipid biomarker panel was established, consisting of ursodeoxycholic acid, PC(O-14:0_20:4), and Cer(d18:1/18:0). This panel exhibited strong diagnostic efficacy, with an AUC of 0.903 for distinguishing early-stage EC patients from healthy controls and an AUC of 0.928 for distinguishing them from atypical endometrial hyperplasia patients. Remarkably, the lipid biomarker panel outperformed the clinically established indicators for EC diagnosis, including HE4, CA125, CA153 and CA199 [65]. Audet-Delage and their research team applied MS-based untargeted metabolomics to analyze pre-operative serum samples from 36 patients with EC and 18 control subjects. Their study revealed an increase in the levels of conjugated lipids, specifically acylcholines, monoacylglycerols, and acylcarnitines in EC cases, while free fatty acids exhibited a decrease. Additionally, they identified an increase in C5 acylcarnitine 2-methyl butyryl carnitine in EC cases [66]. These findings were in line with those reported by Bahado-Singh, who also observed an increase in acylcholines in EC [67].

Some papers suggest acylcholines might enhance the penetration of estradiol into tissues, potentially contributing to endometrial carcinogenesis [63], explaining why multiple research articles have described their dysregulation. On the other hand, acyl-carnitines are 14-carbon fatty acids linked to a carboxylate through an ester bond and play vital roles in mitochondrial fatty acid oxidation [68]. They are known to be enriched in hypoxic tissues and have previously been associated with the biochemistry of breast cancer [68,69,70]. Other lipid metabolites upregulated in EC include monoacylglycerols, which result from the enzymatic breakdown of triacylglycerols and diacylglycerols [66,68]. These glycerides are ultimately metabolized by monoacylglycerol lipase into free fatty acids, a group of lipid metabolites that are downregulated in EC. The potential downregulation of the monoacylglycerol lipase enzyme in EC could theoretically account for these observed findings [66,71].

In a cross-sectional diagnostic accuracy study, Paraskevaidi and colleagues used 342 plasma samples from women with EC, 68 samples of atypical hyperplasia, and 242 healthy controls to demonstrate that spectroscopy has the capability to detect EC with 87% sensitivity and 78% specificity [72]. Notably, the diagnostic accuracy was most pronounced for Type I EC and atypical hyperplasia, with sensitivities of 91% and 100%, and specificities of 81% and 88%, respectively [72]. In a study conducted in 2022 by Arda Düz and team, HR-MAS NMR spectroscopy was employed on a cohort of 17 EC tumor samples and 18 samples of healthy endometrial tissue. The aim was to validate a distinct metabolomic profile associated with EC. The findings of this study suggest elevated levels of lactate, glucose, choline, and various amino acids in EC tumor tissue compared to healthy endometrial tissue. These results closely align with and reinforce the findings reported in several of the studies presented earlier [73].

**Table 1 cancers-16-00185-t001:** Overview of the most relevant metabolomics biomarkers for EC diagnosis.

Metabolite	Group	Platform	Sample Type	Function and Relevance
PC C40:1, PC C42:0, PC44:5 Acylcarnitine C16 Hydroxysphingomyelins [60]	Phospholipids Conjugated lipids Sphingolipid	Electrospray ionization-tandem mass spectrometry	Serum	Related with cell membrane synthesis and transport of fatty acids for B-oxidation
PCs Phosphatidylethanolamine (PE) Phosphatidylinositol (PI) Phosphatidylglycerol (PG) Linoleic acid Glutamine, phenylalanine [61]	Phospholipids Polyunsaturated carboxylic acid Amino acids	UPLC-ESI-TOF-MS In-vitro assays	Tumor and non-tumor tissue samples	Related with cell membrane synthesis, RNA transcription, etc.
PC Malate Asparagine [62]	Phospholipids Dicarboxylic acid Amino acids	NMR	Cervicovaginal Fluid	Related with cell membrane synthesis, protein synthesis, and NADH transport for energy production.
Ursodeoxycholic acid PC(O-14:0_20:4) SM(d18:0/18:0) Cer(d18:1/18:0) HexCer(d18:1/18:1) [63]	Steroid acids Phospholipids Sphingolipid	UHPLC-MS/MS (Lipidomics)	Serum	Pro-inflamatory capacities, de-novo synthesis of ceramides, cell survival and transduction.
PC C14:2 PC C38:1 [65]	Carnitine (Conjugated lipid)	NMR and MS	Serum	Fatty acid transport
Octenoylcarnitine Linoleic acid Stearic acid Valine [68]	Conjugated lipids Polyunsaturated carboxylic acid Saturated monobasic acid Amino acids	GC-MS	Serum	Fatty acid transport, tumor growth, inhibition of tumor growth (downregulated in EC), protein synthesis.
6-keto-PGF1 PA(37:4) LysoPC(20:1) PS(36:0) [71]	Prostaglandin Phospholipids	UPLC-MS (Lipidomics)	Serum	6-keto-PGF1 is a prostaglandin derivative, which can promote tumor growth.
Serine Glutamic acid Phenylalanine Glyceraldehyde-3-phopsphate [58]	Amino acids Sugar	GC-MS	Serum	Protein synthesis, ROS buffering and metabolites of anaerobic glycolysis. (Warburg effect)
Stearamide Monoolein Hypoxanthine 1,2-dihexadecanoyl-sn-glycerol [74]	Endocannabinoids Purine derivative Amino acid derivative	LC-ESI-QTOF-MS/MS	Tumor tissue	Endocannabinoid system regulates cell proliferation, differentiation and survival. Migration of endometrial cells, as well.

Another application for diagnostic biomarkers is differential diagnosis. Endometrial polyps (EP) and endometrial hyperplasia (EH) are lesions very closely related to EC and can eventually develop into it, however, when detected, have to be treated differently. In a study by Yan and colleagues, nontargeted lipidomic analysis was conducted on serum samples from 326 patients with endometrial diseases and 225 healthy volunteers. Through a combination of multivariate and univariate analyses, they successfully identified and validated six, eight, and seven potential biomarkers in the sera of patients with EP, EC, and EH, respectively. Leveraging a logistic regression algorithm and receiver operating characteristic (ROC) curve analysis, a biomarker panel consisting of four specific EP biomarkers (6-keto-PGF1α, PA(37:4), LysoPC(20:1), and PS(36:0)) demonstrated strong classification and diagnostic capabilities in distinguishing EP from EC or EH. For distinguishing EP from EC, this biomarker panel had an AUC of 0.915, with a sensitivity of 100% and a specificity of 72.41%. Likewise, when distinguishing EP from EH, the AUC reached 1.000, with a sensitivity of 100% and a specificity of 100%. Notably, these two diagnostic models exhibited robust diagnostic performance in the validation set as well [74].

However, not only lipid-derivatives have been identified as prospective biomarkers, various amino acids and their derivatives have been proposed as potential diagnostic biomarkers for EC. Amino acids are essential for protein synthesis and are pivotal for sustaining the survival of cancer cells. Moreover, amino acids have the ability to modulate the redox equilibrium and have been associated with the epigenetic and immune regulatory functions within cancer cells [75,76]. In a cornerstone study in the field, Troisi and colleagues developed a unique metabolic signature that is able to accurately diagnose EC based on serine, glutamic acid, phenylalanine and glyceraldehyde 3-phosphate serum concentrations [60]. Using machine learning, they were able to successfully generate a metabolic signature using serum samples from 691 gynecological surgery patients. These samples were divided in three different groups: Training (90 samples), test (38 samples), and validation (563 samples). Afterwards, the signature was tested against a new set of 871 serum samples taken from women with unknown EC status, to evaluate efficiency, accuracy, and overall accuracy of their model, showing an error rate of less than 5% in identifying EC. In another study performed by our group, we utilized an LC-ESI-QTOF-MS/MS platform to reveal a distinct metabolomic signature that has the potential to characterize endometrioid endometrial carcinoma. Our findings implicated the endocannabinoid system as a potential contributor to the pathogenesis of EC. We observed differences in the metabolomic profiles between surface EC and the invasive front within the myometrium, suggesting a potential role of purine metabolism in tumor myometrial invasion [77].

### 4.2. Metabolomics for EC Prognosis and Disease Progression Tracking

Another possibility is identifying key metabolites or developing metabolic signatures that allow accurate prognoses. Prognostic instruments are essential for preoperative risk stratification of patients, enabling informed treatment recommendations, and tailored planning while preventing both under- and over-treatment [78].

In a 2021 paper, Skorupa and colleagues investigated the tissue metabolomic characteristics associated with EC grades. Metabolic profiles were generated from a cohort of 64 patients, including 14 with grade 1 (G1) EC, 33 with grade 2 (G2) EC, and 17 with grade 3 (G3) EC, and these profiles were compared to those from ten patients with benign disorders. The results of this study demonstrated notable changes in metabolite levels. Across all EC grades, in comparison to non-transformed tissue, there were increased levels of valine, isoleucine, leucine, hypotaurine, serine, lysine, ethanolamine, and choline, while levels of creatine, creatinine, glutathione, ascorbate, glutamate, phosphoethanolamine, and scyllo-inositol decreased. Moreover, elevated levels of taurine were detected in both G1 and G2 tumors compared to control tissues. G1 and G3 tumors exhibited increased levels of glycine, N-acetyl compounds, and lactate. Specifically, G1 tumors were characterized by increased dimethyl sulfone and phosphocholine, as well as decreased glycerophosphocholine and glutamine levels. G2 and G3 tumors were distinguished by decreased myo-inositol levels. Additionally, G3 tumors displayed elevated 3-hydroxybutyrate, alanine, and betaine levels. The differences between G1 and G3 malignancies were primarily associated with disruptions in phosphoethanolamine and phosphocholine biosynthesis, inositol metabolism, betaine metabolism, serine metabolism, and glycine metabolism [79].

Audet-Delage and colleagues identified 98 metabolites that exhibited differential expression between Type I and Type II endometrial cells (EC). Among these, 30 metabolites demonstrated higher expression in Type I EC, while 68 exhibited lower expression. Notably, two of the most promising biomarkers were identified: bradykinin, which displayed a 2.7-fold increase in Type I EC (fold-change = 2.70, *p* = 0.003), and heme, which exhibited a 4.5-fold increase in Type II EC [66].

In a previous report from our group, we conducted a metabolomic analysis involving 31 patients with EC, comprising 20 cases of endometrial endometrioid carcinomas (EECs) and 11 cases of serous carcinomas (SCs). Employing multivariate statistical techniques, we discerned 232 metabolites exhibiting significant differences between the SC and EEC patient cohorts. It is noteworthy that the majority of these identified metabolites (89.2%) belonged to the lipid category and exhibited reduced levels in SCs as compared to EECs. In addition to lipids, we also observed variations in metabolites related to amino acids and purine nucleotides, including 2-oxo-4-methylthiobutanoic acid synthesized by the enzyme acireductone dioxygenase 1 (ADI1). Notably, these metabolites displayed higher levels in SCs. To delve deeper into the role of ADI1 in SC, we examined protein expression levels of ADI1 in 96 EC cases (67 EECs and 29 SCs), demonstrating that ADI1 expression was significantly elevated in SCs compared to EECs. Furthermore, we found that ADI1 mRNA levels were higher in ECs with p53 abnormalities compared to those with wild-type p53 tumors. Moreover, our analysis unveiled a statistically significant negative correlation between elevated ADI1 mRNA levels and overall survival, as well as progression-free survival in EEC patients [37]. Alternatively, lipid-derived biomarkers that have been postulated to have prognostic value include picolinic acid, vaccenic acid, phosphatidic acid, arachidonic acid, 13Z-docosenamide, UDP-N-acetyl-d-galactosamine, 1-palmitoyl-2-linoleoyl, inosine, palmitic amide, gleamide, linoleic acid, phosphatidylserine, phosphatidylinositol, and various glycerophosphocholines. Picolinic acid, a byproduct of the kynurenine pathway, is notably downregulated in cases of EC, consistent with its recognized anti-tumoral properties. Conversely, UDP-N-acetyl-D-galactosamine and arachidonic acid exhibit upregulation in advanced stages of EC [63,68].

In another report, Strand and colleagues aimed to investigate potential links between specific metabolic patterns and the characteristics of aggressive disease as well as reduced survival rates among patients from a Norwegian cohort. In this report, the researchers studied 20 patients with EC who had short survival and matched them based on histology and FIGO staging with 20 patients who had long survival. They employed a multiplex system that included 183 metabolites, which were later determined with LC-MS, to differentiate between short and long survival cases. From this project, they extracted a novel metabolite signature associated with survival, with an AUC ranging from 0.820 to 0.965 (*p* ≤ 0.001). Notably, methionine sulfoxide was linked to poor survival rates in these patients. Furthermore, in a subgroup of patients who underwent preoperative contrast-enhanced computed tomography, the researchers observed correlations between selected metabolites and estimated parameters of abdominal fat distribution. These metabolic signatures hold promise for predicting prognosis and could serve as valuable supplements when assessing patient phenotypes and exploring metabolic pathways related to EC progression [80]. Table 2 summarizes the studies described in this section.

## 5. Conclusions

In this review, we aimed to soundly describe the fundamental aspects of EC and metabolomics in order to set a framework that allowed the reader to clearly understand the current state of this young field. Despite the small body of literature, this study includes the latest and most interesting advancements in the field. The data here presented underscores the potential exhibited by various metabolites, mainly lipids and fatty acid derivatives, but also amino acids and hormones, as new EC biomarkers for detection, prognosis, and even treatment monitoring. Despite the very encouraging findings within existing literature, the evidence for clinical translation is still insufficient, since most identified biomarkers have failed to compete against existing clinical tests. In this context, further research is essential to establish a biomarker as a clinically approved test, requiring its confirmation and validation using a substantial number of specimens. Although metabolomics cannot currently be used as a standalone tool for diagnosis, the accumulated research experience and the ongoing exploration of the metabolome ensure that there will be no shortage of newly discovered biomarker metabolites in the future. This is particularly relevant as metabolomics, when combined with minimally invasive sampling techniques holds great potential for delivering clinically relevant biomarkers that could eventually become part of routine clinical practice.

Taking into account the need for sensitive and more affordable diagnosis and prognosis tools, future research efforts should revolve around integrating multi-omics data and generating multi-center datasets that can firmly advocate for the integration of metabolic biomarkers in endometrial cancer management. In the era of personalized medicine, the inherent high-throughput capabilities of metabolomics render it an excellent choice to assess EC treatment efficacy or to opt for alternative therapies during the course of the treatment. Leveraging advancements in artificial intelligence and machine learning techniques will determine the final outcome of these efforts.

## Figures and Tables

**Figure 1 cancers-16-00185-f001:**
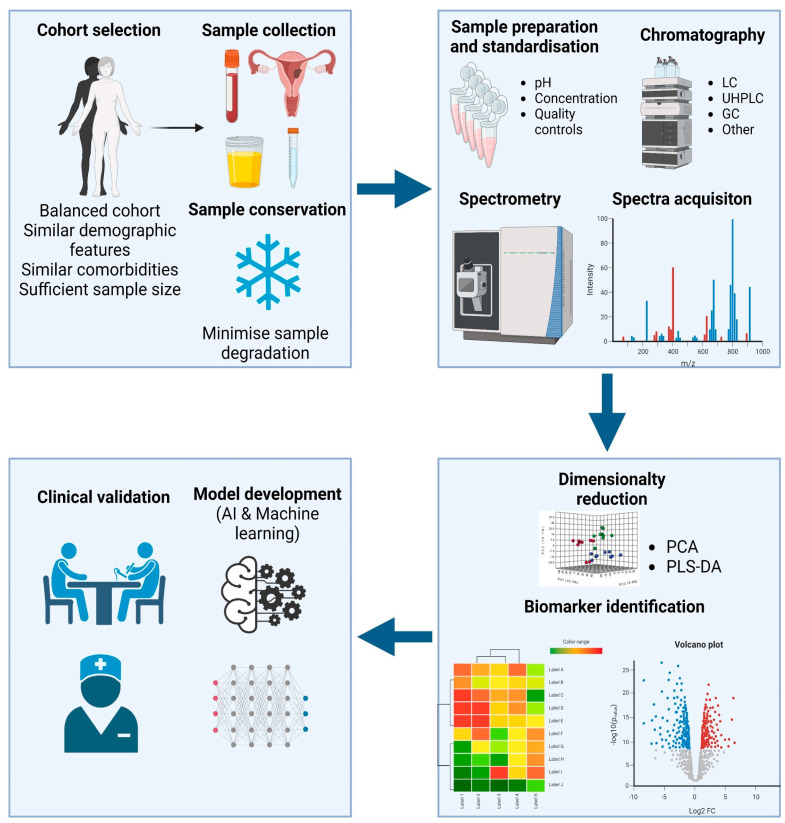
Diagram depicting the typical workflow followed in a biomarker discovery process. Adequate cohort selection, as well as appropriate sample collection and conservation are fundamental to avoid identification of false or irrelevant biomarkers. The most commonly employed platforms are mass spectrometry based. After spectra acquisition, statistical tests are run to discover biomarkers. These targets are finally employed to develop machine learning models and/or validated in a clinical context. Created with Biorender.com.

**Table 2 cancers-16-00185-t002:** Overview of the most relevant metabolomics biomarkers for EC prognosis, staging and disease tracking.

Metabolite	Group	Platform	Sample Type	Function and Relevance
Increased: Valine, Isoleucine, Leucine, Hypotaurine, serine, lysine, ethanolamine, choline. Decreased: Creatine, creatinine, glutathione, ascorbate, glutamate, PE and PC [76]	Amino acids Phospholipids	High resolution magic angle spinning (HR-MAS) proton spectroscopy (NMR)	Tumor tissue	PE and PC are identified as the two most differential biomarkers. They intervene in cell proliferation and metabolism. Amino acidic variation may depend on protein synthesis, ROS buffering, etc.
Bile acids Bradykinin Ceramides Glycine, Cystathionine Heme [64]	Steroid acids Polypeptide Lipid Amino acids Iron-contaning porphyrin	UPLC-MS	Serum	Pro-inflamatory capacities, fatty acid transport, cell signaling, synthesis of cysteine, proteinogenesis, etc.
2-oxo-4-methylthiobutanoic acid [35]	Purine nucleotide	LC-MS/MS	Tumor tissue sample	Increased migration and invasion capabilities.
Methionine sulfoxide SM-C20:2 PC-aa-C36:5 Spermine [77]	Amino acids Phospholipid Phospholipid Polyamine	LC-MS/MS	Serum	Methionine sulfoxide is involved in cell oxidation buffering and biological ageing. SM and PC are involved in cell proliferation and fatty acid distribution. Spermine is involved in cell metabolism.

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
