# Peer review of "Metabolomic-Based Approaches for Endometrial Cancer Diagnosis and Prognosis: A Review"

_cancers, 2023, doi:10.3390/cancers16010185_

Round 1

Reviewer 1 Report

Comments and Suggestions for Authors

This is a well written paper. The authors cover the important topics and clearly understand the topic.  

This is a really good paper

Author Response

We appreciate your positive feedback and encouragement regarding our work. No particular issues or suggestions for modifications or enhancements were identified. We will consider their acknowledgment of the strength and effectiveness of our approach

Reviewer 2 Report

Comments and Suggestions for Authors

Very interesting and valuable article

Author Response

We truly appreciate your feedback and encouraging remarks concerning our submission. You did not raise any particular concerns necessitating modifications or improvements.

Reviewer 3 Report

Comments and Suggestions for Authors

I read with great interest the present Manuscript which falls within the aim of the Journal. In my honest opinion, the topic is interesting enough to attract the readers’ attention. Methodology is accurate and conclusions are supported by the data analysis. Nevertheless, authors should clarify some points and improve the discussion citing relevant and novel key articles about the topic. For all those reasons, I suggested performing the minor revisions.

·      Inclusion and exclusion criteria should be clarified in a Methods section. For example, Authors may specify whether they followed a Flow-diagram in the article selection process;

·      Authors may hypothesize whether specific metabolomic signatures would be predictive of the outcomes of a Fertility-Sparing technique in endometrial cancer surgery. Please consider: “Ronsini C, Mosca L, Iavarone I, et al. Oncological outcomes in fertility-sparing treatment in stage IA-G2 endometrial cancer. Front Oncol. 2022;12:965029. Published 2022 Sep 16. doi:10.3389/fonc.2022.965029”;

·      Please mention novel key evidence about the topic, citing the role of spectroscopy – among the metabolomic techniques in endometrial neoplasms –, and consider: “Arda Düz S, Mumcu A, DoÄŸan B, et al. Metabolomic analysis of endometrial cancer by high-resolution magic angle spinning NMR spectroscopy. Arch Gynecol Obstet. 2022;306(6):2155-2166. doi:10.1007/s00404-022-06587-0”;

·      Authors may discuss the role of liquid biopsy to assess the metabolomic profile of endometrial cancer, similarly to other contexts as endometriosis. Please see: “Ronsini C, Fumiento P, Iavarone I, Greco PF, Cobellis L, De Franciscis P. Liquid Biopsy in Endometriosis: A Systematic Review. Int J Mol Sci. 2023;24(7):6116. Published 2023 Mar 24. doi:10.3390/ijms24076116”;

·      The Authors have not adequately highlighted the strength and limitations of the study. I suggest better specifying those points.

Author Response

We appreciate your thorough assessment of our manuscript. Your positive evaluation of the manuscript's alignment with the journal's scope and the intriguing nature of the topic is encouraging. Your acknowledgment of the accuracy in methodology and the support of conclusions by data analysis is valued.

Regarding the points needing clarification and the recommendation for enhancing the discussion with citations from pertinent and recent literature, we are grateful for your constructive feedback. We have diligently addressed these areas in our revision to ensure a more comprehensive and refined discussion, incorporating relevant and novel references.

Below, you'll find a detailed response addressing the suggested improvements point by point:

  • Inclusion and exclusion criteria should be clarified in a Methods section. For example, Authors may specify whether they followed a Flow-diagram in the article selection process;

In response to the suggestions provided, we have incorporated a dedicated section on Materials and Methods in the revised manuscript. This section comprehensively outlines the inclusion and exclusion criteria. We believe these additions address the concerns raised and enhance the clarity of our methodology

  • Authors may hypothesize whether specific metabolomic signatures would be predictive of the outcomes of a Fertility-Sparing technique in endometrial cancer surgery. Please consider: “Ronsini C, Mosca L, Iavarone I, et al. Oncological outcomes in fertility-sparing treatment in stage IA-G2 endometrial cancer. Front Oncol. 2022;12:965029. Published 2022 Sep 16. doi:10.3389/fonc.2022.965029”;

We appreciate Reviewer's insightful suggestion regarding the potential use of specific metabolomic signatures in predicting outcomes following Fertility-Sparing techniques in endometrial cancer surgery. We have taken into account the reference provided and have integrated the relevant information into our manuscript (page 2, lines 86-90).

  • Please mention novel key evidence about the topic, citing the role of spectroscopy – among the metabolomic techniques in endometrial neoplasms –, and consider: “Arda Düz S, Mumcu A, DoÄŸan B, et al. Metabolomic analysis of endometrial cancer by high-resolution magic angle spinning NMR spectroscopy. Arch Gynecol Obstet. 2022;306(6):2155-2166. doi:10.1007/s00404-022-06587-0”;

Thank you for your valuable suggestion. We have incorporated the requested information into our manuscript. Specifically, on page 10, lines 384-390 we have included a reference to the study by Arda Düz et al., emphasizing the role of spectroscopy among metabolomic techniques in endometrial neoplasms. We appreciate your guidance in enhancing our manuscript.

  • Authors may discuss the role of liquid biopsy to assess the metabolomic profile of endometrial cancer, similarly to other contexts as endometriosis. Please see: “Ronsini C, Fumiento P, Iavarone I, Greco PF, Cobellis L, De Franciscis P. Liquid Biopsy in Endometriosis: A Systematic Review. Int J Mol Sci. 2023;24(7):6116. Published 2023 Mar 24. doi:10.3390/ijms24076116”;

We genuinely appreciate your valuable input and the provided insightful reference. While exploring the role of liquid biopsy in assessing metabolomic profiles in both endometriosis and endometrial cancer is of significant interest, it is essential to highlight the distinct nature and origins of these two pathologies. Endometriosis and endometrial cancer, originate from fundamentally different biological contexts. The intricacies of endometriosis, being an ectopic tissue condition, and endometrial cancer, a neoplastic transformation, underscore the need for separate and specialized discussions.

Considering the unique characteristics and clinical implications of each condition, we believe that a separate review article dedicated to elucidating the distinct metabolomic aspects of endometriosis would be particularly valuable. It would allow for a comprehensive exploration of the specific metabolomic signatures and their relevance in understanding the pathophysiology and diagnosis of endometriosis.

  • The Authors have not adequately highlighted the strength and limitations of the study. I suggest better specifying those points.

We have addressed the issue by enhancing the Conclusion section of our manuscript. The strengths and limitations of the study are now better specified, providing a more comprehensive understanding for readers. We appreciate your guidance in refining our work

We deeply appreciate the time and dedication you've devoted to evaluating our work, which has enriched the quality and robustness of our research.